# Tunable directional photon scattering from a pair of superconducting qubits

Elena S. Redchenko ©[1] ✉, Alexander V. Poshakinskiy[2], Riya Sett[1], Martin Žemlička[1], Alexander N. Poddubny ©[3] & Johannes M. Fink[1] ✉

The ability to control the direction of scattered light is crucial to provide flexibility and scalability for a wide range of on-chip applications, such as integrated photonics, quantum information processing, and nonlinear optics. Tunable directionality can be achieved by applying external magnetic fields that modify optical selection rules, by using nonlinear effects, or interactions with vibrations. However, these approaches are less suitable to control microwave photon propagation inside integrated superconducting quantum devices. Here, we demonstrate on-demand tunable directional scattering based on two periodically modulated transmon qubits coupled to a transmission line at a fixed distance. By changing the relative phase between the modulation tones, we realize unidirectional forward or backward photon scattering. Such an in-situ switchable mirror represents a versatile tool for intra- and inter-chip microwave photonic processors. In the future, a lattice of qubits can be used to realize topological circuits that exhibit strong nonreciprocity or chirality.

One of the simplest ways to realize directional light scattering relies on the Kerker effect[1,2]. It is based on the interference between different multipole components of scattered light, for example electric and magnetic dipoles, and has been demonstrated for Si nanoparticles[3–5]. However, the nanoparticle scattering pattern is fixed after fabrication and dictated by its shape. Tunable light routing is typically enabled by an external magnetic field that leads to the Zeeman splitting of optical transitions for clockwise- and counter-clockwise- propagating photons[6] or a modification of optical selection rules[7]. In the optical domain, the routing can be reversed also without changing the magnetic field by flipping the spin of the atom[8]. Such structures are now actively studied in the domain of chiral quantum optics[9,10]. Tunable directional scattering can also be achieved by using moving boundary conditions[11,12]. For example, the trembling of a small particle with only an electric dipole resonance can induce a magnetic dipole resonance[11], which in turn provides directional scattering in analogy to the Kerker effect. Several compact nonmagnetic realizations of nonreciprocal devices using Raman and Brillouin scattering[13–17] have been presented for optical frequencies.

Tunable directional interactions are also highly desired for superconducting quantum circuits in the microwave spectral range. For instance, isolators and circulators are commonly used for superconducting quantum computing to protect fragile qubits states. Cascaded photon processing in the chiral setup is also beneficial for the creation of complex entangled quantum states of qubits[18]. However, it is difficult to directly extend existing approaches for visible light to microwave photons. For example, the classical Kerker approach is not applicable to a typical transmon qubit that behaves just as an electric dipole[19], without magnetic dipole resonances. Devices, based on an external magnetic field[20], are often bulky and always require additional shielding to protect superconducting qubits. While there exist demonstrations of directionality in parametrically driven, compact mechanical systems[21–23], integration with superconducting circuitry is challenging due to limited bandwidth and tunability. Thus, there is a need for flexible to use on-chip microwave photon routers, which do not require strong magnetic fields or moving mechanical parts.

Here, our goal is to demonstrate an easy-to-fabricate circuit providing frequency and directionality tunable photon scattering with the

[1]Institute of Science and Technology Austria, 3400 Klosterneuburg, Austria. [2]Ioffe Institute, St. Petersburg 194021, Russia. [3]Weizmann Institute of Science, Rehovot 7610001, Israel. ✉e-mail: elena.redchenko@ist.ac.at; johannes.fink@ist.ac.at

minimum number of components required. Our approach is based on the sinusoidal time-modulation of the qubit frequency[24–28], which is a standard technique in circuit and waveguide QED. The modulated qubit strongly coupled to a waveguide can be mapped onto the problem of light scattering from the trembling mirror[11,29,30]. By altering the relative phase $\alpha$ between the modulation tones of two qubits, we change the effective phase shift between the scattered sidebands resulting in different interference patterns for forward and backward scattering as schematically shown in Fig. 1a. Here, we do not focus on the elastic scattering nonreciprocity[25,26,31] or directional emission from the initial qubit state[32,33] but on the switching between forward and backward inelastic coherent scattering. Thus, although elastically (Rayleigh) scattered radiation remains almost unaffected, we gain the flexibility to choose the frequency of the scattered photons.

## Results

### Experimental implementation

We fabricate the sample with two transmon qubits coupled to a 1D coplanar transmission line separated by $d = 5$ mm as shown in Fig. 1b. The maximum frequency of the $|0\rangle \rightarrow |1\rangle$ transition is 9.129 (9.577) GHz for Qubit 1(2). We tune both qubits to $\omega_0/(2\pi) = 6.129$ GHz corresponding to an effective distance of $d = \lambda/4$, with $\lambda$ the wavelength of photons at $\omega_0$, using bias coils mounted on top of the sample box. Currents for the periodic frequency modulation are applied via on-chip bias lines inductively coupled to the SQUID loops as shown in Fig. 1c. Working away from a sweet spot with a close to linear flux

dispersion lowers the required modulation currents. Both ports of the transmission line are connected to separate microwave in- and output lines to measure reflection and transmission spectra simultaneously.

Firstly, we characterize the qubits individually at $\omega_0$ where $d = \lambda/4$ using a weak resonant probe tone and measuring the coherently and elastically scattered radiation, i.e., at the same frequency. We determine the normalized transmission spectrum of each qubit shown in Fig. 1d, e and find the radiative decay rates to be $\Gamma_1/(2\pi) \approx 4.4$ MHz and the dephasing rates of $\Gamma_2/(2\pi) \approx 3.9(4.3)$ MHz for Qubit 1 (2)[19]. The corresponding pure dephasing rates are $\Gamma_\varphi/(2\pi) \approx 1.7(2.1)$ MHz dominated by flux noise due to the relatively high flux dispersion at this bias point. Here, we assume other sources of decay to be small in comparison[34].

An applied sinusoidal bias current makes the qubit resonance frequency tremble in time and the coherent transmission amplitude is then given by

$$t_0 = 1 + \sum_{n=-\infty}^{\infty} \frac{i\Gamma_1/2}{\omega_0 + n\Omega - \omega - i\Gamma_2} J_n^2\left(\frac{A_m}{\Omega}\right), \tag{1}$$

where $J_n(\frac{A_m}{\Omega})$ are Bessel functions of the first kind, $A_m$ is the modulation amplitude in frequency units, and $\Omega$ is the modulation frequency. We measure the normalized transmission spectrum $|t_0|^2$ as a function of modulation frequency $\Omega$ as shown in Fig. 2a. For the fixed signal amplitude at the AWG output $A_V = 50$ mV$_{pp}$, the system undergoes a

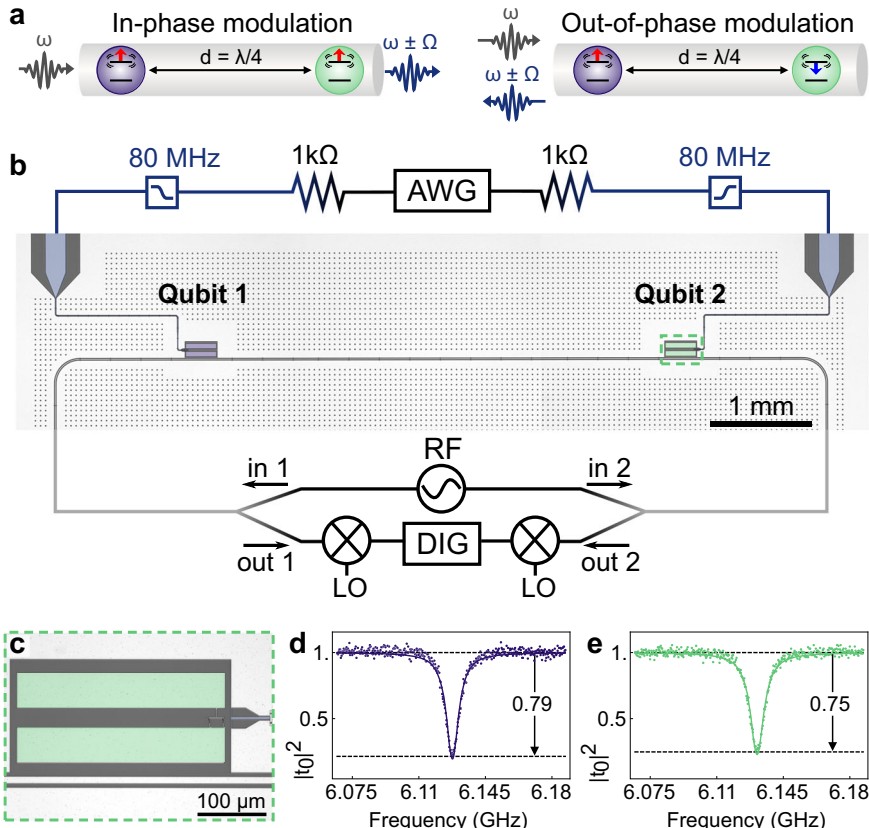

**Fig. 1 | Experimental realization. a** Schematic showing the scattering direction of the $\omega \pm \Omega$ component for in-phase (up-up) and out-of-phase (up-down) modulation of the qubits' transition frequencies $\omega$. **b** Optical microscope image and simplified experimental setup. Two transmon qubits are capacitively coupled to a 50 Ω transmission line, and each qubit has a local flux bias line connected to an arbitrary waveform generator channel (AWG), which is used to generate a sinusoidal wave with an amplitude $A_V$ that is filtered with a 80 MHz low-pass filter and

applied to ground via a 1 kΩ resistor. We use an RF source, analog down-conversion and digitization (DIG) to back out the scattering parameters of the device cooled to 10 mK. **c** Enlarged view of Qubit 2 and local flux bias line inductively coupled to the qubit SQUID. **d**, **e** Individually measured and normalized transmission spectra $|t_0|^2$ of elastically scattered radiation from Qubit 1(2) with fit to theory (solid line).

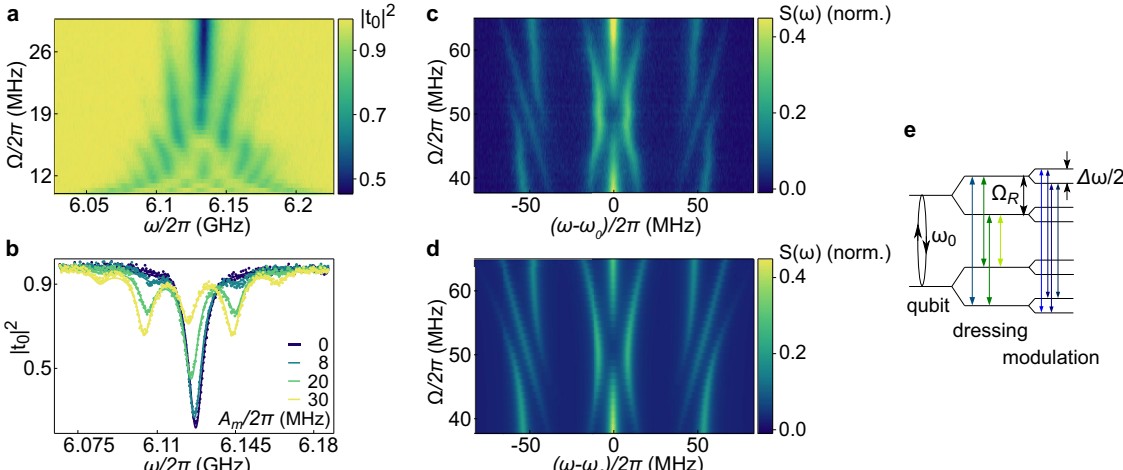

**Fig. 2 | Singe qubit properties. a** Normalized transmission spectrum $|t_0|^2$ of qubit 1 measured as a function of the modulation frequency $\Omega$ and the probe frequency $\omega$ at the fixed modulation amplitude $A_V = 50$ mV$_{pp}$. **b** Measured $|t_0|^2$ of the modulated qubit with $\Omega/(2\pi) = 20$ MHz for different $A_m$ and fits to Eq. (1) (solid lines). **c** Measured resonance fluorescence emission spectrum of qubit 1 as a function of the modulation frequency $\Omega$ and detuning of the detected inelastically scattered radiation from the drive applied at $\omega_0$ for a Rabi frequency $\Omega_R/(2\pi) = 52$ MHz and modulation amplitude $A_m = 0.2\,\Omega_R$. **d** Theoretically predicted Mollow spectrum in the presence of frequency modulation for the same parameters. **e** Level splitting schematics of the dressed and modulated qubit, which explains the origin of the observed nested Mollow triplets at $\Omega = \Omega_R$.

transition from the strong ($A_m/2 > \Omega$) to the weak ($A_m/2 < \Omega$) modulation regime, which is also referred to as Landau-Zener-Stückelberg-Majorana spectroscopy[24,28]. We fit similarly measured data to Eq. (1) for both qubits individually, as shown in Fig. 2b. This yields the dependence $A_m(\Omega)$ for a fixed $A_V$ as well as $A_m(A_V)$ for a fixed $\Omega$, which is approximately linear. Examples for both are shown in the Supplementary Methods I.

### Modulated Mollow resonance fluorescence

One of the hallmark characteristics of quantum two-level systems is the observation of the incoherent resonance fluorescence spectrum taking the form of a Mollow triplet for an applied resonant drive of sufficient power $\Omega_R > \Gamma_1$[19]. Here, we observe this effect for a frequency-modulated qubit with $A_m = 0.2\,\Omega_R$ and Rabi frequency $\Omega_R/(2\pi) = 52$ MHz. The measured power spectral density (PSD) as a function of the modulation frequency $\Omega$ is shown in Fig. 2c and the corresponding theory in Fig. 2d. Dressing with the drive leads to the well known emission spectrum with three maxima at $\omega_0$ and $\omega_0 \pm \Omega_R$. However, the additional frequency modulation leads to the formation of avoided crossings at $\Omega = \Omega_R$, which can be qualitatively interpreted as a formation of nested Mollow triplets following the level scheme shown in Fig. 2e. Specifically, each of the levels of the original Mollow triplet is split into two levels due to the modulation. Next, the photon transitions between the split levels lead to the formation of additional Mollow triplets. For example, the transition from the original triplet having the largest energy, and shown by the thick vertical blue arrow, is transformed by the modulation into three distinct transition energies shown by the thin blue lines. The observed splitting between the outermost transitions of the inner Mollow triplets for $\Omega = \Omega_R$ is equal to $\Delta\omega/(2\pi) \approx 20$ MHz, in excellent agreement with the numerical calculation.

Similar formations of nested Mollow triplets in the electron spin-noise spectrum have been predicted for the conditions of electron paramagnetic resonance when the electron is subject to a the time-modulated magnetic field[35], but have so far not been observed directly to the best of our knowledge.

### Directional scattering

Now we consider the system of two qubits both tuned to $\omega_0$ and located at a distance $\lambda/4$. For any odd multiple of $\lambda/4$ a single resonant

microwave tone drives the two qubits with opposite phase, which leads to a coherent exchange interaction mediated by virtual photons[36] forming a coupled two-qubit molecule[37,38]. In the absence of modulation, the backscattering is suppressed by destructive interference[18], while the interference for forward scattering is constructive. The addition of frequency modulation of both qubits results in nontrivial interference conditions for the Stokes and anti-Stokes sidebands, as shown in the insets of Fig. 3. For these measurements we chose $\Omega/(2\pi) = 20$ MHz and $A_m/(2\pi) = 20$ MHz to fully resolve a small number of sidebands (see Supplementary Methods II). The blue and green arrows correspond to the incident light (dashed) and the inelastically scattered light (solid) at $\pm 20$ MHz from the first and second qubit, respectively. If the two modulation tones are in phase ($\alpha = 0$), illustrated in the insets of Fig. 3a and c by red arrows inside the qubits (up-up), the device continues to scatter light only in the forward direction since its symmetry is not modified by the modulation. Accordingly, we observe sidebands mostly scattered forward and almost fully suppressed in backscattering (dashed circles in panels a and c). However, if the modulation has a phase difference of $\alpha = \pi$, the situation is reversed. This is illustrated by the blue arrows inside the second qubit (up-down) in the insets of Fig. 3b and d, corresponding to an additional phase factor of $-1$. While the inelastic backscattering is now highly likely as shown in Fig. 3d, the sidebands scattered forward from the first qubit destructively interfere with the ones scattered from the second one due to the additional phase shift and thus preventing forward scattering as shown in Fig. 3b.

In order to better illustrate the phase and detuning dependence of the interference conditions we extract the coherent scattering power of the Stokes component over the full range of $\alpha$ and for finite detuning from the qubit resonances at $\omega_0$. For this measurement the detection frequency is always detuned by the chosen modulation frequency $\Omega/(2\pi) = 20$ MHz from the probe tone at frequency $\omega$. Here, we detect both the transmitted and reflected scattered Stokes quadratures with the two channels of the digitizer simultaneously for $A_m/(2\pi) = 30$ MHz. The corresponding power in transmission and reflection is shown in Fig. 4a and b. We observe resonances at probe frequencies $\omega_0$, $\omega_0 \pm \Omega$, and $\omega_0 - 2\Omega$, and their overall dependence on $\alpha$ is clearly pronounced and opposite in sign for forward and backward scattering. The measured FWHM bandwidth of directional photon scattering centered at $\omega_0 - \Omega/2$ is around 25 MHz as shown in

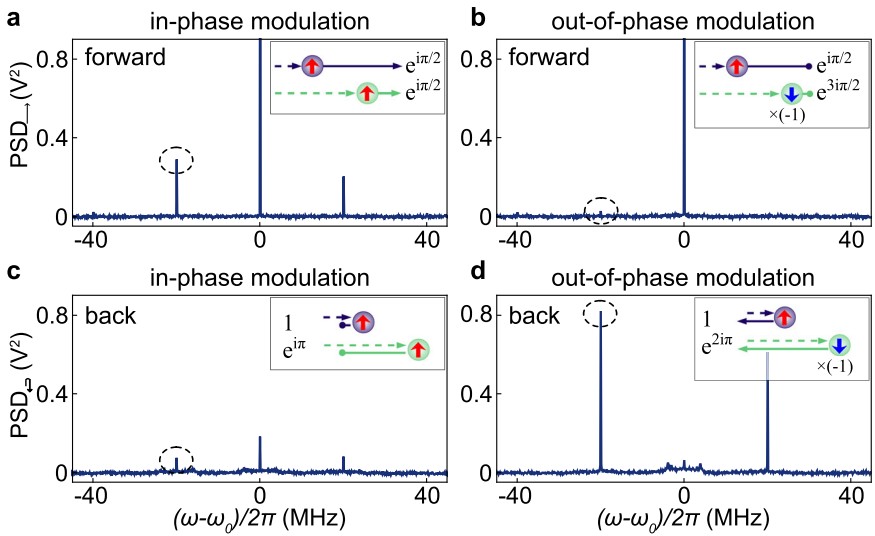

**Fig. 3 | Resonance fluorescence spectra.** Power spectral density (PSD) measured in transmission (**a**, **b**) and reflection (**c**, **d**) at the digitizer for in-phase $\alpha = 0$ (**a**, **c**) and out-of-phase $\alpha = \pi$ (**b**, **d**) modulation. The Stokes components are highlighted with dashed circles. Scattering schematics are shown as insets where blue (green) arrows represent the light scattered from qubit 1 (2) at $\omega_0 \pm \Omega$ leading to constructive interference in **a** and **d** or destructive interference in **b** and **c**. Full Rayleigh peak heights are 1.9 and 1.6 $V^2$ for the chosen settings in **a** and **b**.

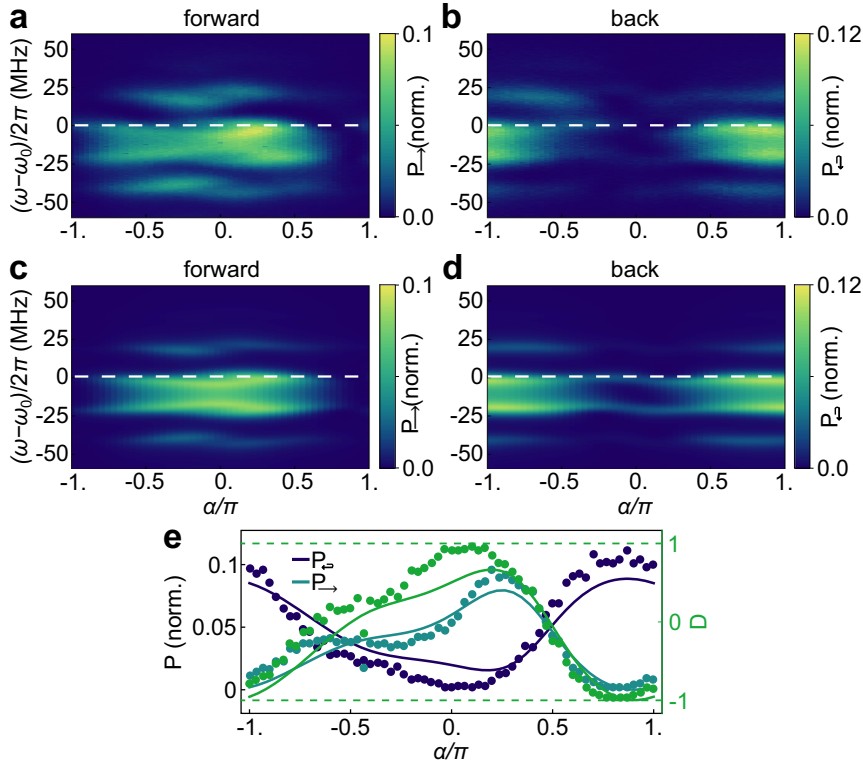

**Fig. 4 | Coherent inelastic scattering spectrum of the Stokes component.** **a**, **b** Measured and normalized Stokes power (squared quadratures) as a function of probe frequency detuning $\omega - \omega_0$ and relative phase between modulation tones $\alpha$ measured in transmission (reflection) at a fixed modulation amplitude $A_m/(2\pi) = 30$ MHz and modulation frequency $\Omega/(2\pi) = 20$ MHz. **c**, **d** Theoretically predicted transmission (reflection) spectrum on the same scale. For better agreement, here we include small frequency shifts of $-0.6$ and $-0.8$ MHz between the two qubits and $\omega_0$. **e** Coherent inelastic scattering as a function of $\alpha$ measured at $\Omega/(2\pi) = -20$ MHz from the probe frequency indicated with dashed lines in **a**–**d** (points) and theory (solid lines). Scattering directivity $D$ is shown in green.

Supplementary Methods III. These experimental results are in very good agreement with the theoretical model shown in Fig. 4c, d, see Methods for details.

The measured dependence of the scattering parameters on the phase difference $\alpha$ as well as the directivity $D = (P_\rightarrow - P_\leftarrow)/(P_\rightarrow + P_\leftarrow)$ is shown in Fig. 4e for the probe frequency on-resonance with the qubit

frequencies $\omega = \omega_0$ (dashed white lines in panels a-d) together with theory. This shows phase selective control to enter the regimes when light is mostly scattered back ($D < 0$), forward ($D > 0$), or symmetrically in both directions ($D = 0$). The measured directivity demonstrates high diode efficiency that can be set continuously between $D = \pm 0.96$.

## Discussion

In this work, we have explored a novel regime of light-matter interaction that is characterized by an interplay between nonlinear photon scattering, photon-mediated qubit-qubit interaction and sideband generation via parametric modulation. The studied physics also applies to other implementations, e.g., based on atomic scatterers or mechanically modulated devices. Our result adds to the growing interest of time-modulated qubits and interactions that have, e.g., been suggested for faster 2-qubit gate implementations[39], entanglement stabilization[40] and already been used for controlled photon release in photonic cluster state generation[41].

The demonstrated high level of scattering directivity of ±0.96 could also become useful as part of an on-chip microwave photon router that can be switched on-demand between scattering photons backward, forward, or symmetrically in both directions using collective interference. The suppression strength of the inelastically scattered light forward at $\alpha/\pi = \pm 1$ and backward at $\alpha/\pi = 0$ ($|10 \log P_{\to(\leftarrow)}(\alpha/\pi = 0) - 10 \log P_{\to(\leftarrow)}(\alpha/\pi = \pm 1)|$) of up to 16 dB can be modified with the modulation amplitude and the signal frequency can be shifted and fine-tuned in-situ by changing the modulation frequency, e.g., to address multiple frequency bands in analogy to frequency division multiplexing in classical communication. Moreover, a larger range of frequency bands can be accessed by working at odd multiples of the $\lambda/4$ boundary condition.

Previously realized single photon routers[42] relied on a qubit in the linear regime, which naturally limits the operation of the device to low powers $(\Omega_R/\Gamma_1)^2 \ll 1$. In contrast, our calculations indicate that scattering remains directional up to intermediate drive powers $(\Omega_R/\Gamma_1)^2 \lesssim 9$ beyond which the inelastic scattering is fully suppressed, see Supplementary Methods IV. With a bandwidth of 25 MHz it is also fully compatible with modern superconducting quantum computing devices[32,43] but one of the main limitations of the current device is its high insertion loss compared to state-of-the-art routers and switches[42,44,45]. We estimate that this insertion loss can be improved to as low as $-2.1$ dB by reducing pure dephasing and by suppressing unwanted frequency bands by means of a structured waveguide forming a bandpass[41,46], see Supplementary Methods II.

Besides the demonstrated high scattering directionality at the relative phase $\alpha/\pi = \pm 1$, our system also exhibits the characteristics of a microwave isolator at the relative phase $\alpha/\pi = \pm 0.4$ based on the traveling-wave modulation[47], which might be further enhanced with optimized device parameters or by extending the principle to a larger number of qubits, details can be found in the Supplementary Methods V. In the future, such an extension to multiple nodes, see theory in Methods, might be used to realize topologically protected states[48], as a part of a hardware implementation of Gottesman-Kitaev-Preskill codes[49], or to route microwave radiation for the realization of chiral networks[9,50].

## Methods

### Calculation of scattering spectra

In this section, we present the general approach to calculate photon scattering from an array of qubits with time-modulated resonance frequencies. Such a device is characterized by the following effective non-Hermitian Hamiltonian[51]:

$$
\begin{aligned}
H(t) = &\sum_j [\omega_0^{(j)}(t) - i\Gamma_2^{(j)}]\sigma_j^\dagger \sigma_j - \frac{i\Gamma_1}{2} \sum_{j,k} e^{i\varphi|j-k|}\sigma_j^\dagger \sigma_k \\
&+ \frac{\Omega_R}{2} \sum_j (\sigma_j^\dagger e^{i\varphi j - i\omega t} + \text{H.c.}).
\end{aligned}
\tag{2}
$$

Here, $\sigma_j$ are the raising operators, $\Gamma_1$ is the (radiative) relaxation rate between the $|1\rangle$ and $|0\rangle$ qubit states, $\Gamma_2^{(j)}$ is the decay rate of the coherence between the $|1\rangle$ and $|0\rangle$ states, $\varphi = \omega_0 d/c$ is the phase

gained by light traveling between the qubits with propagation velocity $c$. The Rabi frequency $\Omega_R$ quantifies the incident wave amplitude and

$$
\omega_0^{(j)}(t) = \omega_0 + A_m \cos(\Omega t + \alpha_j)
\tag{3}
$$

are the time-dependent qubit resonance frequencies. The Hamiltonian Eq. (2) assumes the usual rotating wave and Markovian approximations. Here, we are interested in the case of weak coherent driving where the wavefunction can be approximately written as

$$
\psi = |0\rangle + \sum_j p_j \sigma_j^\dagger |1\rangle.
\tag{4}
$$

The amplitudes $p_j$ describe the coherence between the ground and excited states and can be found from the following effective Schrödinger equation:

$$
i\frac{d}{dt}p_j(t) = [\omega_0^{(j)}(t) - i\Gamma_2^{(j)}]p_j - \frac{i\Gamma_1}{2} \sum_k e^{i\varphi|j-k|}p_k + \frac{\Omega_R}{2} e^{i\varphi j - i\omega t}.
\tag{5}
$$

It is convenient to seek the solution in the form

$$
p_j(t) = \sum_{n=-\infty}^{\infty} p_j^{(n)} e^{-i(\omega + n\Omega)t},
\tag{6}
$$

so that the amplitudes $p_j^{(n)}$ are determined by the linear system[52]

$$
\begin{aligned}
(\omega + n\Omega)p_j^{(n)} = &(\omega_0 - i\Gamma_2^{(j)})p_j^{(n)} + \frac{A_m}{2}(e^{i\alpha_j}p_j^{(n-1)} + e^{i\alpha_j}p_j^{(n+1)}) \\
&- \frac{i\Gamma_1}{2} \sum_k e^{i\varphi|j-k|}p_k^{(n)} + \frac{\Omega_R}{2} e^{i\varphi j}\delta_{m,0}.
\end{aligned}
\tag{7}
$$

After the amplitudes $p_j^{(n)}$ have been found numerically, we calculate the coefficients $r^{(n)}$ and $t^{(n)}$

$$
r^{(n)} = -\frac{i\Gamma_1}{\Omega_R} \sum_j e^{i\varphi j}p_j^{(n)},
\tag{8}
$$

$$
t^{(n)} = \delta_{n,0} - \frac{i\Gamma_1}{\Omega_R} \sum_j e^{-i\varphi j}p_j^{(n)},
\tag{9}
$$

that describe the backward (forward) scattering process with the frequency change $\omega \to \omega + n\Omega$. In the general case, the system of equations (7) is to be solved numerically. However, it is possible to obtain an analytical solution in the particular case of a single qubit[53]. In this case we find

$$
p^{(n)} = \frac{\Omega_R}{2} \sum_{n'=-\infty}^{\infty} \frac{J_{n'-n}(A_m/\Omega)J_{n'}(A_m/\Omega)}{\omega + n'\Omega - \omega_0 + i\Gamma_2}.
\tag{10}
$$

For elastic scattering ($n = 0$) Eq. (10) leads to Eq. (1) in the main text.

### Resonance fluorescence of the time-modulated device

Here we describe the procedure to calculate the nested Mollow triplets shown in Fig. 2. The state of the qubit can be conveniently represented

as vector $\boldsymbol{S}$ of the spin 1/2, where $|1\rangle$ and $|0\rangle$ states correspond to $S_z = 1/2$ and $-1/2$, respectively. The dynamics $\boldsymbol{S}(t)$ is governed by the Bloch equation that reads

$$\frac{d\boldsymbol{S}}{dt} = \boldsymbol{S} \times \widetilde{\boldsymbol{\Omega}}(t) - \boldsymbol{\Gamma}(\boldsymbol{S} - \boldsymbol{S}_0) \tag{11}$$

where

$$\widetilde{\boldsymbol{\Omega}}(t) = [\Omega_R \cos\omega t, \Omega_R \sin\omega t, \omega_0 + \Delta\omega\cos(\Omega t + \alpha)] \tag{12}$$

is the time-dependent effective magnetic field, $\boldsymbol{S}_0 = [0, 0, -1/2]$ is the equilibrium spin, and the spin relaxation term reads $\boldsymbol{\Gamma}(\boldsymbol{S} - \boldsymbol{S}_0) \equiv [\Gamma_2 S_x, \Gamma_2 S_y, \Gamma_1(S_z + 1/2)]$. The emission spectrum is determined by the spin correlation function

$$I(\omega) \propto \text{Re} \int_0^\infty dt e^{-i\omega\tau} \langle\langle S_+(t + \tau) S_-(t)\rangle\rangle_t, \tag{13}$$

where $S_\pm = S_x \pm iS_y$ and the double angular brackets denote averaging over the absolute time $t$. Equation (13) establishes the correspondence between the emission spectrum in the considered quantum optics problem and the electron spin-noise spectrum in the conditions of electron paramagnetic resonance, when the electron is subject to two magnetic fields, a constant one and an oscillating one[35].

In the theory of magnetic resonance, the standard trick to solve Eq. (11) analytically is to switch to a reference frame rotating around the $z$ axis with the drive frequency $\omega$. There, the spin dynamics is governed by the same Eq. (11) but $\widetilde{\boldsymbol{\Omega}}(t)$ shall be replaced with

$$\boldsymbol{\Omega}'(t) = [\Omega_R, 0, \omega_0 - \omega + \Delta\omega\cos\Omega t]. \tag{14}$$

In the absence of modulation, $\Delta\omega = 0$, the effective magnetic field $\boldsymbol{\Omega}'(t)$ would be constant and its amplitude

$$\Omega_R' = \sqrt{\Omega_R^2 + (\omega_0 - \omega)^2} \tag{15}$$

would determine the splitting in the conventional Mollow triplet.

The presence of modulation can be accounted for by repeating the trick and switching to yet another frame rotating with frequency $\Omega_R'$ with respect to the previous one. There, $\boldsymbol{\Omega}'(t)$ is replaced with

$$\boldsymbol{\Omega}'' = \frac{\Omega_R \Delta\omega}{2\Omega_R'^2}[\omega - \omega_0, 0, \Omega_R] + \left(1 - \frac{\Omega}{\Omega_R'}\right)[\Omega_R, 0, \omega_0 - \omega], \tag{16}$$

where we neglected all oscillating terms, since they average to zero. The amplitude of $\boldsymbol{\Omega}''$ determines the splitting of the nested Mollow triplet

$$\Omega_R'' = \sqrt{\left(\frac{\Omega_R \Delta\omega}{2\Omega_R'}\right)^2 + (\Omega_R' - \Omega)^2}. \tag{17}$$

Returning back to the initial reference frame, we obtain nine possible emission frequencies

$$\omega_{p,q} = \omega + p\Omega_R' + q\Omega_R'', \tag{18}$$

where $p, q = 0, \pm 1$ enumerate the components of the two nested Mollow triplets. In the above analytical solution, we used twice the rotating wave approximation, which is valid provided $\Delta\omega \ll \Omega_R \ll \omega_0$.

### Data normalization
We normalize the transmission spectra $|t_0|^2$ shown in Fig. 1d, e and Fig. 2a, b by dividing the background transmission coefficient

$|t_0|^2 = |t|^2/|t_{bg}|^2$. Here, $|t_{bg}|^2$ is measured with both qubits tuned out of the frequency range of interest, and $|t|^2$ is measured with the qubit tuned to the desired frequency. This method normalizes the gain in the system and compensates for the frequency-dependent transmission properties of the drive and detection lines.

The power spectral density of the measured resonance fluorescence spectrum $S(\omega)$ shown in Fig. 2c, as well as the coherent inelastic scattering spectra shown in Fig. 4a, b were scaled to the numerically predicted value. The latter relies on the qubit parameters extracted from the transmission measurements, the chosen modulation frequency, and the independently calibrated modulation amplitude.

### Data availability
All datasets and analysis files used in this study are available at https://doi.org/10.5281/zenodo.7858567.

### Code availability
Code used in this study is available at https://doi.org/10.5281/zenodo.7858567.

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

## Acknowledgements

The authors thank W.D. Oliver for discussions, L. Drmic and P. Zielinski for software development, and the MIBA workshop and the IST nano-fabrication facility for technical support. This work was supported by the Austrian Science Fund (FWF) through BeyondC (F7105) and IST Austria. E.R. is the recipient of a DOC fellowship of the Austrian Academy of Sciences at IST Austria. J.M.F. and M.Z. acknowledge support from the European Research Council under grant agreement No 758053 (ERC StG QUNNECT) and a NOMIS foundation research grant. The work of A.N.P. and A.V.P. has been supported by the Russian Science Foundation under the grant No 20-12-00194.

## Author contributions

E.S.R. designed and fabricated the samples, worked on the setup, and performed the measurements. A.V.P. and A.N.P. developed the theory and together with E.S.R conducted the data analysis. M.Z. contributed to building the measurement setup and R.S. to the resonance fluorescence data acquisition code. E.S.R. and A.N.P. wrote the manuscript with contributions from all authors. J.M.F. supervised this work.

## Competing interests

The authors declare no competing interests.
