## [Peer review file · Nature Communications]

REVIEWER COMMENTS

Reviewer #1 (Remarks to the Author):

I am satisfied with the authors' reply to the referees. I find the level of novelty of the manuscript appropriate for Nature Communications. However, before recommending publication, I would like to ask the authors to carefully consider my comments below.

1. Page 4, when stating "The measured FWHM bandwidth of directional photon scattering is around 25 MHz", please give an argument why this is to be expected, given the linewidth of each qubit is about 4 MHz.

2. P. 4, discussion, the authors quote both a directivity of 0.96 and a suppression strength of 18.5 dB, where does the second number come from? If derived from the first, maybe better to leave it out for clarity.

3. Page 4, "our system also exhibits the characteristics of a microwave isolator at the relative phase $\alpha/\pi = \pm 0.3$ ", this statement should be clarified also in the main text and not only in the SM. In any case, this concept requires more clarifications. The scattering matrix that the authors intend to compare to (S5) is constructed assuming that incident radiation is at the qubit frequency and scattered radiation is frequency-converted. I assume isolation in this case means that radiation is not converted but rather scattered at the original frequency (or converted to other frequencies). This scattering could be detrimental to the applications discussed at the end of the manuscript, and it is not obvious to me that it can be suppressed by any combination of passive, reciprocal microwave components (dissipative or not).

4. Supplementary Discussion I, I understand how the authors arrive at an estimate for the remaining insertion loss, by assuming that emission into all undesired sidebands can be suppressed "using a structured waveguide forming a bandpass". However, their calculation is not a proper treatment of scattering from quantum emitters in the presence of a structured bath, which would require appropriate modifications to the master equation. In principle, one cannot exclude that the emission properties of the emitters could be modified by the additional structure of the bath, in such a way that the intended conversion process is affected, and the reported figure worsened. Do the authors have arguments to exclude that? If not, their estimate should be regarded as lower bound for the insertion loss, to be further verified.

5. Suppl Fig. S5, It'd be nice to have the y axis range starting from 0.

We thank the Reviewer for their generally positive response and we would like to thank them for their constructive comments. Below, we respond to all of the concerns point by point and also highlight all implemented manuscript changes with red color.

Reply to reviewer #1

I am satisfied with the authors' reply to the referees. I find the level of novelty of the manuscript appropriate for Nature Communications. However, before recommending publication, I would like to ask the authors to carefully consider my comments below.

1. Page 4, when stating "The measured FWHM bandwidth of directional photon scattering is around 25 MHz", please give an argument why this is to be expected, given the linewidth of each qubit is about 4 MHz.

Unfortunately, there is no analytical expression letting to predict this bandwidth (see Methods. Calculation of scattering spectra), however, from our numerical simulations of two frequency-modulated qubits, we see that the bandwidth of each of the inelastically scattered components at ω_0 , $\omega_0 \pm \Omega$, and $\omega_0 - 2\Omega$ is slightly broader than the sum of qubits' linewidths (see Supplementary Discussion II). That phenomenologically leads to the expected bandwidth of each component of at least ~ 8 MHz. In our case of modulation frequency $\Omega/2\pi = 20$ MHz, the numerically predicted and experimentally observed broadening leads to the merging of ω_0 and $\omega_0 - \Omega$ components and FWHM bandwidth of directional photon scattering of 25 MHz is centered around $\omega_0 - \Omega/2$. **In the revised version, we have additionally specified the central frequency of the 25 MHz wide directionally scattered light in the main text.**

2. P. 4, discussion, the authors quote both a directivity of 0.96 and a suppression strength of 18.5 dB, where does the second number come from? If derived from the first, maybe better to leave it out for clarity.

The directivity in our manuscript is defined as a diode efficiency $D = (P_{\rightarrow} - P_{\leftarrow}) / (P_{\rightarrow} + P_{\leftarrow})$ and the suppression strength refers to the difference in the power of inelastically scattered light forward (backward) at the relative phases $\alpha/\pi = 0$ and $\alpha/\pi = \pm 1$: $|10\log[P_{\rightarrow(\leftarrow)}(\alpha/\pi = 0)] - 10\log[P_{\rightarrow(\leftarrow)}(\alpha/\pi = \pm 1)]|$. As one can see, the suppression strength is a phase-independent parameter that can be changed by adjusting the modulation strength. **We have added a definition of the suppression strength value in the discussion part of the manuscript.**

3. Page 4, "our system also exhibits the characteristics of a microwave isolator at the relative phase $\alpha/\pi = \pm 0.3$ ", this statement should be clarified also in the main text and not only in the SM. In any case, this concept requires more clarifications. The scattering matrix that the authors intend to compare to (S5) is constructed assuming that incident radiation is at the qubit frequency and scattered radiation is frequency-converted. I assume isolation in this case means that radiation is not converted but rather scattered at the original frequency (or converted to other frequencies). This scattering could be detrimental to the applications discussed at the end of the manuscript, and it is not obvious to me that it can be suppressed by any combination of passive, reciprocal microwave components (dissipative or not).

We agree that the isolation of the inelastically scattered components might not be of interest for future applications. Nonetheless, our system also exhibits isolator behavior for elastically scattered light at the qubit frequency at relative phases $\alpha/\pi = \pm 0.4$, where the shown scattering matrix is applied. The observed non-reciprocity of our microwave device is based on the travelling-wave modulation similar to the electrically induced non-reciprocity at optical frequencies (H. Lira et. al., PRL 109, 033901, 2012).

Thus, we have updated Supplementary Fig. S5 with the measurements and theory for the elastically scattered light, have added further concept explanation in the discussion part of the main text and to supplementary discussion, and updated the values for the phase α/π , isolation, and insertion loss.

4. Supplementary Discussion I, I understand how the authors arrive at an estimate for the remaining insertion loss, by assuming that emission into all undesired sidebands can be suppressed “using a structured waveguide forming a bandpass”. However, their calculation is not a proper treatment of scattering from quantum emitters in the presence of a structured bath, which would require appropriate modifications to the master equation. In principle, one cannot exclude that the emission properties of the emitters could be modified by the additional structure of the bath, in such a way that the intended conversion process is affected, and the reported figure worsened. Do the authors have arguments to exclude that? If not, their estimate should be regarded as lower bound for the insertion loss, to be further verified.

We fully agree with the Referee that the presence of a structured waveguide might modify the emission process. The rigorous calculation of the insertion loss, in that case, would require modifications to the effective Schrödinger equation, Eq. (5) in Methods. It falls, however, beyond the scope of this paper, as it is a complex theoretical task on its own. In our calculations, we followed two main assumptions: that the emission into all undesired sidebands is suppressed and that the ratio between emission into the allowed inelastic and elastic scattering channels stays the same which gives a slightly higher bound. Additionally, as we mentioned in the Supplementary Discussion I, we did not optimize the relative phase α/π that could reduce the insertion loss also in the case of the modified emission process.

We have further clarified these assumptions and the limitations of our estimation of the subsequent loss reduction.

5. Suppl Fig. S5, It’d be nice to have the y axis range starting from 0.

The Y-axis range in Supplementary Fig. S5 is indeed starting from 0. We thank the Referee for pointing out the unfortunate choice of the y-axis ticks. As suggested, we changed them in the revised version of the manuscript.

REVIEWERS' COMMENTS

Reviewer #1 (Remarks to the Author):

I am satisfied with the authors' replies to my comments and the changes they made in the manuscript and supplementary information. I recommend publication.